# The Expression Kinetics and Immunogenicity of Lipid Nanoparticles Delivering Plasmid DNA and mRNA in Mice

**DOI:** 10.3390/vaccines11101580

**Published:** 2023-10-11

**Authors:** Wanyue Zhang, Annabelle Pfeifle, Casey Lansdell, Grant Frahm, Jonathon Cecillon, Levi Tamming, Caroline Gravel, Jun Gao, Sathya N. Thulasi Raman, Lisheng Wang, Simon Sauve, Michael Rosu-Myles, Xuguang Li, Michael J. W. Johnston

**Affiliations:** 1Centre for Oncology, Radiopharmaceuticals and Research, Biologic and Radiopharmaceutical Drugs Directorate, Health Products and Food Branch, Health Canada and World Health Organization Collaborating Center for Standardization and Evaluation of Biologicals, Ottawa, ON K1A 0K9, Canada; wanyue.zhang@hc-sc.gc.ca (W.Z.); annabelle.pfeifle@hc-sc.gc.ca (A.P.);; 2Department of Biochemistry, Microbiology and Immunology, Faculty of Medicine, University of Ottawa, Ottawa, ON K1H 8M5, Canada; lisheng.wang@uottawa.ca; 3Centre for Vaccines, Clinical Trials and Biostatistics, Biologic and Radiopharmaceutical Drugs Directorate, Health Products and Food Branch, Health Canada and World Health Organization Collaborating Center for Standardization and Evaluation of Biologicals, Ottawa, ON K1A 0K9, Canada; 4Department of Chemistry, Carleton University, Ottawa, ON K1S 5B6, Canada

**Keywords:** lipid nanoparticles, DNA vaccines, mRNA vaccines, bioluminescent imaging, in vivo imaging, lipid composition, luciferase, ionizable lipids, expression kinetics, immunogenicity

## Abstract

In recent years, lipid nanoparticles (LNPs) have emerged as a revolutionary technology for vaccine delivery. LNPs serve as an integral component of mRNA vaccines by protecting and transporting the mRNA payload into host cells. Despite their prominence in mRNA vaccines, there remains a notable gap in our understanding of the potential application of LNPs for the delivery of DNA vaccines. In this study, we sought to investigate the suitability of leading LNP formulations for the delivery of plasmid DNA (pDNA). In addition, we aimed to explore key differences in the properties of popular LNP formulations when delivering either mRNA or DNA. To address these questions, we compared three leading LNP formulations encapsulating mRNA- or pDNA-encoding firefly luciferase based on potency, expression kinetics, biodistribution, and immunogenicity. Following intramuscular injection in mice, we determined that RNA-LNPs formulated with either SM-102 or ALC-0315 lipids were the most potent (all *p*-values < 0.01) and immunogenic (all *p*-values < 0.05), while DNA-LNPs formulated with SM-102 or ALC-0315 demonstrated the longest duration of signal. Additionally, all LNP formulations were found to induce expression in the liver that was proportional to the signal at the injection site (SM102: r = 0.8787, *p* < 0.0001; ALC0315: r = 0.9012, *p* < 0.0001; KC2: r = 0.9343, *p* < 0.0001). Overall, this study provides important insights into the differences between leading LNP formulations and their applicability to DNA- and RNA-based vaccinations.

## 1. Introduction

Since the start of the COVID-19 pandemic, over 90 million doses of lipid nanoparticle (LNP)-based mRNA vaccines have been administered in Canada alone [1]. The success of these immunization programs has brought nucleic acid technologies to the forefront of vaccine research. LNPs are a critical component of these vaccine platforms, presenting several key advantages for the delivery of nucleic acids.

First, encapsulation in an LNP protects nucleic acids from degradation and improves stability in biological fluids [2,3,4,5]. Second, LNPs improve cellular uptake, leading to increased expression of the target antigen, which may contribute to increased immunogenicity [5,6,7,8,9]. Third, nucleic acid vaccines delivered by LNPs do not require additional adjuvants for immune activation [10,11]. Finally, when compared to delivery by viral vectors, LNP-based vaccines are more easily produced and do not induce anti-vector immunity that may hinder vaccine efficacy [12].

To date, all approved LNP-based vaccines have been designed to encapsulate modified mRNA. Despite the success of this technology, there are several limitations associated with mRNA vaccines, including poor stability, cold storage requirements, and high production costs [13,14,15,16,17]. In comparison, plasmid DNA (pDNA) vaccines are more thermostable and less susceptible to degradation [17,18,19,20,21]. We have observed that DNA-LNPs maintain transfection potency better than mRNA-LNPs after one week at 37 °C (Appendix A). Moreover, DNA vaccines are less expensive to produce, store, and transport than mRNA vaccines [22,23]. LNP-based DNA vaccines therefore have the potential to alleviate a number of issues inherent to mRNA vaccine technology, which could improve suitability for more wide-spread use.

Recently, the first DNA vaccine was approved in India for the prevention of COVID-19 [24]. Many other DNA vaccines in clinical and preclinical development are administered using intradermal injection or specialized instruments such as gene guns or needle-free injectors [25,26]. Considering that these techniques may not be easily translated into human vaccination programs, LNPs offer a safe and reliable alternative for delivering DNA vaccines by intramuscular injection [6,27,28]. Despite these advantages, there are currently no DNA vaccines approved or undergoing clinical trials that are delivered by LNPs. One study by Mucker and colleagues demonstrated that LNP encapsulation increased the neutralizing antibody titres induced by DNA vaccines for Andes virus and Zika virus [6]. In addition, a study by Algarni et al. demonstrated that a DNA-LNP vaccine formulated with the ionizable lipid DLin-KC2-DMA (KC2), resulted in greater antigen expression than the leading DLin-MC3-DMA (MC3)-formulated particle when administered intramuscularly [29]. These studies provide promising insights into the potential of DNA-LNP vaccines as an alternative platform for vaccine development.

To our knowledge, the potency, expression kinetics, biodistribution, and immunogenicity conferred by DNA-LNP vaccines formulated with the ionizable lipid KC2 or the lipid formulations utilized in clinical COVID-19 vaccines (SM-102 and ALC-0315) have not yet been investigated [30,31]. This study aims to bridge a significant knowledge gap regarding the efficacy and characteristics of DNA-LNP vaccines formulated with different lipid components. Development of DNA vaccines can be accelerated by employing LNPs that have already demonstrated safety and efficacy in the context of mRNA vaccines. Therefore, comparisons of mRNA and DNA formulated with the same particle should be explored. To address these questions, we generated three LNP formulations encapsulating mRNA- or pDNA-encoding firefly luciferase. We characterized each particle and then evaluated the potency of these particles in vitro. Furthermore, we evaluated the intensity and duration of the luminescence signal in vivo at both the injection site and distally, in the liver. Finally, the anti-luciferase antibody response induced by each formulation was evaluated as a surrogate for vaccine immunogenicity.

## 2. Materials and Methods

### 2.1. Preparation of pDNA- and mRNA-Encoding Luciferase

mRNA-encoding firefly luciferase (mRNA-Luc) was purchased from TriLink Biotechnologies (San Diego, CA, USA) with the substitution of N1-Methylpseudouridine for uridine.

To generate the pDNA-encoding firefly luciferase gene (pVAX1-Luc), the Luc gene from the pcDNA3-Luc plasmid (Addgene, Watertown, MA, USA) was cloned into the pVAX1 plasmid vector (ThermoFisher, Ottawa, ON, Canada) using HindIII-HF and XbaI restriction enzymes (New England Biolabs, Whitby, ON, Canada). pVAX1-Luc was transformed into NEB^®^ 10-beta Competent *E. coli* (High Efficiency) (New England Biolabs, Ipswich, MA, USA)according to the manufacturer’s protocol). Large-scale amplifications of pVAX1-Luc were generated using the EndoFree QIAGEN Plasmid Giga Kit (Montreal, QC, Canada) according to the manufacturer’s instructions.

### 2.2. Lipid Nanoparticle (LNP) Generation

To generate the LNPs, an aqueous phase containing pDNA, mRNA, or buffer alone was combined with an organic phase containing lipids in ethanol via rapid microfluidic mixing using a NanoAssemblr BT™ (Precision Nanosystems, Inc., Vancouver, BC, Canada) as described previously [32]. Samples containing no nucleic acid or pDNA were prepared with 25 mM acetate buffer (pH 4.0), while samples containing mRNA were prepared with 50 mM citrate buffer (pH 4.0). The ethanol phase was composed of ionizable lipid: phospholipid: cholesterol: PEG-lipid at the molar ratios listed in Table 1, using a concentration of 15 mM total lipid. After mixing, the particles were dialyzed against a 1000-fold volume of phosphate-buffered saline (PBS) using a 10 k MWCO cassette (Thermo Fisher, Ottawa, ON, Canada) at 4 °C for 18 h. LNPs were 0.22 μm filtered, then concentrated using an Amicon Ultra 4 10 k MWCO centrifugal concentrator (Millipore Sigma, Burlington, ON, Canada). All particles were made with a polymer amine (N = nitrogen) group to nucleic acid phosphate (P) group (N/P) ratio of 6:1. 

Lipids purchased from MedKoo Biosciences, Inc. (Morrisville, NC, USA) include 2,2-dilinoleyl-4-dimethylaminoethyl-[1,3]-dioxolane (DLin-KC2-DMA), heptadecan-9-yl 8-((2-hydroxyethyl)(6-oxo-6-(undecyloxy)hexyl)amino)octanoate (SM-102), [(4-Hydroxybutyl)azanediyl]di(hexane-6,1-diyl) bis(2-hexyldecanoate) (ALC-0315) and 2-[(polyethylene glycol)-2000]-N,N-ditetradecylacetamide (ALC-0159). Lipids purchased from Avanti Polar Lipids, Inc. (Alabaster, AL, USA) include 1,2-dioleoyl-sn-glycero-3-phosphocholine (18:1 (Δ9-Cis) PC (DOPC)), 1,2-distearoyl-sn-glycero-3-phosphocholine (18:0 PC (DSPC)), 1,2-Dimyristoyl-rac-glycero-3-methoxypolyethylene glycol-2000 (DMG-PEG 2000) and cholesterol (ovine).

### 2.3. Nanoparticle Characterization

Nanoparticle size characterization was performed as described previously [32]. In brief, the particle size was determined by nanoparticle tracking analysis (NTA) (NanoSight, Malvern Panalytical, Westborough, MA, USA) under static conditions. Five separate tracking videos were merged to generate the sizing data. Due to smaller particle size, dynamic light scattering (DLS) (Zetasizer Ultra, Malvern, Panalytical, Westborough, MA, USA) was used to measure LNPs without nucleic acid.

Nucleic acid encapsulation efficiency was measured as previously described [32]. In brief, LNPs were either untreated or disrupted with 1% Triton X-100 (Millipore Sigma, Burlington, ON, Canada) in a 96-well plate before the addition of SYBR™ Gold dye (Thermo Fisher, Ottawa, ON, Canada) to a final concentration of 1 ×. Nucleic acid detected in wells without triton was considered unencapsulated, while nucleic acid detected in wells where triton was added to disrupt the LNP represented total nucleic acid.

### 2.4. In Vitro Transfection Assay

HEK293T cells were cultured in Dulbecco’s Modified Eagle Medium (DMEM) (Thermo Fisher, Ottawa, ON, Canada) supplemented with 10% fetal-bovine serum (FBS). In a 24-well plate, cells were transfected with 500 ng of pDNA or mRNA alone or encapsulated in LNPs, diluted in PBS (Gibco, Thermo Fisher, Ottawa, ON, Canada). RNA- and DNA-transfected cells were incubated at 37 °C, 5% CO_2_ for 16 h or 48 h, respectively, according to the peak transfection time for each payload. The media was then replaced with 200 µL of 1× passive lysis buffer (Promega, Madison, WI, USA) and incubated for 30 min at room temperature on an orbital shaker at 50 rpm. The lysate was centrifuged at 15,000× *g* for 5 min at room temperature, and the cleared lysate was added in triplicate, 100 µL per well, to a white Costar 96-well plate (Corning, Glendale, AZ, USA) followed by 100 µL of room temperature Bright-Glo Reagent (Promega, Madison, WI, USA) prepared according to manufacturer instructions. The luminescence was read within 5 min and normalized to total protein content as measured by a Bicinchoninic Acid (BCA) Kit (MilliporeSigma, Oakville, ON, Canada). The in vitro potency was expressed as relative luminescence units/mg of protein (RLU/mg).

### 2.5. Animal Study

Six-week-old female BALB/c mice were obtained from Charles River, Senneville, Quebec, Canada. All animal procedures were performed in accordance with institutional guidelines and ethical approval was granted by the Animal Care Committee at Health Canada, Ottawa, ON, Canada.

Mice were randomly divided into eight groups, with three to four in each group. At time zero, mice received intramuscular injection into the left and right tibialis anterior muscle, with a total injection volume of 50 µL. The injection contained either 1 µg of mRNA or 25 µg of pDNA encapsulated in LNPs diluted in PBS. An LNP containing no nucleic acid (empty particle) control group was also included in the study and dosed with the equivalent amount of lipid as the DNA-LNP groups. Depilatory cream was applied to the mouse legs under anesthesia to improve visualization prior to initial injection and again 17 days after injection.

Bioluminescence imaging was performed with IVIS Lumina XRMS imaging system (Perkin Elmer, Woodbridge, ON, Canada). At each of the 6 h, 24 h, 48 h, 72 h, 168 h, 336 h, 504 h, and 672 h timepoints after injection, mice were administered intraperitoneally with IVISbrite D-Luciferin Potassium Salt Bioluminescent Substrate (Perkin Elmer) dissolved in PBS at 150 mg/kg. After 15 min, the mice were anesthetized in a chamber with 5% isoflurane at 1.5 L/min flow rate. The mice were placed on the imaging platform while maintained on 2% isoflurane at 500 cc/min flow rate via a nose cone. The mice were imaged 20 min after receiving D-Luciferin using the automatic exposure setting. Bioluminescence values were quantified by measuring photon flux (photons/second) in the region of interest using Living Image software (version 4.7.3) provided by Perkin Elmer.

### 2.6. Enzyme-Linked Immunosorbent Assay (ELISA)

High-binding Nunc Maxisorp™ flat bottom 96-well plates were coated with 100 µL/well of 2 µg/mL of recombinant luciferase (Promega) in PBS. After overnight incubation at 4 °C, plates were washed with PBS containing 0.05% Tween 20 (PBS-T) and blocked with 3% (*w*/*v*) Bovine Serum Albumin (IgG-Free, Protease-Free) (Jackson Immuno Research, West Grove, PA, USA) in PBS-T. After washing, two-fold serial dilutions of the mouse serum in blocking buffer were added for one hour at 37 °C. Plates were washed again, then HRP-conjugated goat anti-mouse IgG (Cytiva, Marlborough, MA, USA), HRP-conjugated goat anti-mouse IgG1 (Jackson Immuno Research, West Grove, PA, USA), or HRP-conjugated goat anti-mouse IgG2a (Jackson Immuno Research) were added at a 1:2000 dilution and incubated for one hour at 37 °C. 75 µL of tetramethylbenzidine (TMB) substrate (Cell Signaling Technology, Danvers, MA, USA) was added after washing and plates were incubated at room temperature for five minutes. The reaction was terminated by addition of 0.16 M sulfuric acid and absorbance was read at 450 nm on a spectrophotometer. Endpoint antibody titers were expressed as the reciprocals of the final detectable dilution, with a cut-off value defined as the mean of all wells containing serum from mice in the empty particle control group plus three times the standard deviation.

### 2.7. Mathematical and Statistical Analyses

Statistical analyses were performed using GraphPad Prism 9 and SAS Enterprise Guide 7.1. The total in vivo signal (total flux^2^) was calculated as area under the curve (AUC) of the fitted signal curves minus the average AUC of the empty particle control group. Statistical differences between groups for in vitro signal, total in vivo signal, and liver and injection site ratio were analyzed using an unpaired two-tailed *t*-test with a significance threshold of *p* < 0.05. The cut-off value for in vivo signal duration was defined as the mean of the signal from the empty particle group plus three times the standard deviation. When comparing the same formulation delivering DNA versus RNA, comparisons between peak signal at the injection site and liver was calculated using a Welch’s (Satterthwaite) *t*-test with a significance threshold of *p* < 0.05. The comparisons between different LNP formulations for DNA and RNA were carried out using analysis of variance (ANOVA). If the result of ANOVA was significant (*p* < 0.05), then the pairwise comparisons of least square mean were carried out using two tailed *t*-test, and the obtained *p*-values were adjusted for multiple comparisons (Bonferroni method). Correlations were analyzed using Pearson correlation. The anti-luciferase IgG endpoint titres and IgG2a:IgG1 ratio were compared using Mann–Whitney two-tailed U-test with a significance threshold of *p* < 0.05. * *p*-value < 0.05, ** *p*-value < 0.01, *** *p*-value < 0.001, **** < 0.0001.

## 3. Results

### 3.1. Characterization of Nanoparticle Formulations

We first generated three LNP formulations containing the ionizable lipids SM-102, ALC-0315, or KC2 (Figure 1, Table 1). The formulations for the ALC-0315- and SM-102-based particles were identical to those used in the approved COVID-19 mRNA vaccines [30,31,33]. To evaluate the suitability of each formulation to deliver both DNA-based and RNA-based vaccines, either 25 µg of pDNA- or 1 µg of mRNA-encoding firefly luciferase was encapsulated in each LNP formulation. The size distributions of the resulting LNPs were analyzed by nanoparticle tracking analysis. The mean particle sizes for the formulations in the present study ranged from 67 to 83 nm (Table 1). Previous reports have demonstrated that the optimal size range for efficient uptake by antigen presenting cells and induction of a robust immune response is between 20 and 200 nm [34,35,36]. Furthermore, all formulations demonstrated high encapsulation efficiencies, ranging between 88 and 99% when evaluated by SYBR™ Gold assay.

### 3.2. Potency of LNP Formulations

#### 3.2.1. SM102-RNA Particles Result in the Highest Luciferase Expression In Vitro

The potency of the RNA- and DNA-LNPs was first evaluated in vitro. HEK293T cells were incubated with LNPs containing firefly luciferase mRNA or pDNA for 16 or 48 h, respectively. For both RNA- and DNA-LNPs, the SM-102 formulation resulted in the highest luciferase expression, likely due to the unique structural characteristics of SM-102. For RNA-LNPs, the ALC-0315 formulation demonstrated the second-best transfection capability. However, when delivering DNA, the KC2 formulation performed better than the ALC-0315 formulation (Figure 2). For both the SM-102 and ALC-0315 formulations, transfection with RNA resulted in higher expression of luciferase than transfection with DNA. Peak transfection occurred at similar levels for the KC2-LNPs regardless of payload.

#### 3.2.2. SM102-RNA and ALC0315-RNA Particles Result in the Highest Luciferase Expression In Vivo

Next, BALB/c mice were injected intramuscularly with the RNA- or DNA-LNP formulations to evaluate the in vivo potency and expression kinetics. Six hours later, mice were injected intraperitoneally with luciferin substrate and imaged using an IVIS Lumina XRMS imaging system to quantify luciferase expression. The imaging was repeated 24 h, 48 h, 72 h, 168 h (7 days), 336 h (14 days), 504 h (21 days), and 672 h (28 days) after the initial LNP injection (Figure 3A).

For all RNA and DNA formulations, maximum luciferase expression was observed at six hours (Figure 3B–D). All three DNA particles yielded similar peak expression levels at the six-hour time point. In comparison, over the complete time course, the total luciferase expression for both the SM102-DNA and ALC0315-DNA groups was significantly higher than the KC2-DNA group. Importantly, no significant difference was observed between the SM102-DNA and ALC0315-DNA groups (Figure 3E).

When delivering an mRNA payload, both SM-102 and ALC-0315 particles resulted in the highest luciferase signal of all groups at the six-hour peak, with no significant difference between the two formulations (Figure 3D,E). The peak intensity of KC2-RNA particles was approximately 20-fold lower than ALC0315-RNA (*p* = 0.002) and SM102-RNA (*p* = 0.0021). This result is consistent with the in vitro results where SM102-RNA and ALC0315-RNA both outperformed KC2-RNA. Over the complete time course, the total luciferase expression was not significantly different between the SM102-RNA and ALC0315-RNA groups; however, both groups resulted in significantly higher expression than the KC2-RNA group (Figure 3E). 

For each formulation, the peak luciferase expression was greater when encapsulating 1 µg of mRNA than 25 µg of pDNA. SM102-RNA and ALC0315-RNA resulted in greater total luciferase expression over the 28-day period. In addition, the total luciferase expression generated by the KC2-RNA group was significantly higher than the KC2-DNA group (Figure 3E). These results suggest that, when comparing mRNA and DNA as LNP payloads, significantly lower doses of mRNA can result in increased protein expression in vivo. Overall, all LNP formulations generated significant luciferase expression when delivering both DNA and mRNA; however, the mRNA payload resulted in more efficient protein expression.

### 3.3. Expression Kinetics: SM102-DNA and ALC0315-DNA Particles Result in the Longest Duration of Protein Expression

Following the peak luciferase expression observed at the six-hour time point, protein expression steadily decreased over the remaining time course for all groups. The KC2-DNA formulation resulted in signal significantly above baseline for the shortest duration of all groups, lasting only 48 h. In comparison, the SM102-DNA and ALC0315-DNA groups were found to have detectable luciferase signal at the final time point of 28 days (Figure 3C, Table 2). SM102-RNA and ALC0315-RNA groups yielded signal for 21 days, while the KC2-RNA formulation lasted a shorter duration of 14 days (Figure 3D, Table 2). Overall, the SM102-DNA and ALC0315-DNA particles resulted in the longest duration of protein expression when compared to all other groups.

### 3.4. Biodistribution: LNPs Result in Protein Expression in the Liver Proportional to the Expression at the Injection Site

Significant luciferase expression was observed in the livers of all mice injected with each of the LNP formulations, with the exception of the KC2-DNA group. The cut-off value for significant luciferase expression was defined as the mean of the signal from the empty particle group plus three times the standard deviation. This finding is in line with previous reports that have also demonstrated antigen expression in the liver following vaccination [30,33]. Here, we found that luciferase expression in the liver of both SM102-RNA and ALC0315-RNA groups lasted for the longest duration of seven days. In comparison, the liver signal in KC2-RNA and ALC0315-DNA mice was present for 72 h, while SM102-DNA mice had liver signal for 48 h (Figure 4A,B, Table 2). Similar to the expression kinetics at the injection site, all groups had the highest liver signal at six hours. ALC0315-RNA and SM102-RNA resulted in the greatest peak liver signal, while the KC2-RNA group was over 50-fold lower than ALC0315-RNA (*p* = 0.0267) (Figure 4A). In addition, both ALC0315-DNA and SM102-DNA resulted in much lower liver signal than their mRNA counterparts at six hours (ALC0315: *p* = 0.0428, SM102: *p* = 0.0093). Similar trends were observed for the total luciferase signal in the liver over the 28-day time course. The total signal was not significantly different between ALC0315-LNPs and SM102-LNPs when both encapsulated the same nucleic acid. Furthermore, both SM102-RNA and ALC0315-RNA particles induced significantly greater total signal than the KC2-RNA group (Figure 4C). Consequently, we sought to determine if the protein expression in the liver was correlated to expression levels at the injection site. When considering all time points with detectable liver signal, the expression at the two sites was strongly correlated for all LNP formulations (SM102: r = 0.8787, *p* < 0.0001; ALC0315: r = 0.9012, *p* < 0.0001; KC2: r = 0.9343, *p* < 0.0001) (Figure 4D). Finally, we determined the ratio of liver signal to injection site signal. The ALC0315 formulation was found to have a significantly higher liver-to-injection site ratio than the SM102 formulation (Figure 4E). Taken together, these results suggest that LNPs lead to significant protein expression in the liver, which is correlated with the expression levels at the injection site and may vary depending on the LNP formulation.

### 3.5. Immunogenicity: SM102-RNA and ALC0315-RNA Particles Result in the Highest Antibody Response with Th1 Bias

Finally, we sought to investigate the luciferase-specific antibody response in the serum 28 days after the initial injection. Despite generating significant luciferase expression, all DNA-LNP groups and the KC2-RNA group did not induce a significant antibody response when compared to mice injected with the negative control LNP (empty particle) (Figure 5A). In comparison, SM102-RNA and ALC0315-RNA both induced significant anti-luciferase antibody titres, with no significant difference in titres between the two groups. Further analysis of the IgG subclass antibody titres revealed that both formulations induced more IgG2a than IgG1, indicating a Th1-biased immune response (Figure 5B). In addition, the antibody titres from all groups were found to moderately correlate with both the 6 h peak luciferase expression (r = 0.5843, *p* = 0.0034) and the total luciferase expression over the 28-day time course (AUC, r = 0.5924, *p* = 0.0029) (Figure 5C,D). In conclusion, the SM102-RNA and ALC0315-RNA groups were found to be the most effective at inducing a robust antibody response after one dose.

## 4. Discussion

In this study, we compared three different LNP formulations delivering an mRNA or pDNA surrogate vaccine based on four key parameters: (1) potency (defined as reporter gene expression), (2) expression kinetics, (3) biodistribution, and (4) immunogenicity. Two of the lipid formulations evaluated in this study are identical to those used in COVID-19 mRNA vaccines, here referred to as SM102- and ALC0315-LNPs. We also included a formulation previously shown to effectively deliver pDNA, referred to as KC2-LNPs [29]. Each of the LNP formulations was used to encapsulate either pDNA- or mRNA-encoding firefly luciferase. Luciferase was used as a representative vaccine antigen as it has been shown previously to be immunogenic and its expression is easily visualized via bioluminescent imaging [37].

First, we evaluated the potency of the LNP formulations in vitro using HEK293T cells. We found that all LNP formulations were effective at delivering DNA, with SM102-DNA > KC2-DNA > ALC0315-DNA. To our knowledge, this is the first demonstration that SM-102- and ALC-0315-based LNPs are effective for in vitro transfection of pDNA. Previous studies have suggested that ionizable lipids with higher pKa values result in a greater quantity of charged ionizable lipids in the endosome leading to improved endosomal release of DNA into the cytoplasm and higher levels of protein expression in vitro [38]. Our results support this conclusion since ionizable lipids with higher pKa values (SM-102 = 6.75 and KC2 = 6.7) were found to have greater reporter gene expression than ALC-0315 (pKa = 6.09). The difference in reporter gene expression between SM102-DNA and KC2-DNA LNPs is likely attributed to SM-102 disrupting the endosomal membrane to a greater extent due to its more cone-like shape [29]. Compared to ALC-0315 and KC2, SM-102 has a more asymmetric tail, which has been shown to have higher transfection efficiency [39]. Interestingly, for RNA-LNPs a different pattern was observed with SM102-RNA > ALC0315-RNA > KC2-RNA suggesting that factors other than pKa, such as LNP morphology, inner structure, and RNA microenvironment, may account for the observed differences [40]. In another recent publication, the authors observed several lipid structure activity relationships that correlated with improved protein expression, including number of carbons on the lipid tails on the ester side and the effect of carbon spacing on the disulfide arm of the lipids [41]. The results of our study are consistent with a recent publication by Escalona-Rayo et al. that demonstrated that SM102-RNA particles resulted in greater in vitro protein expression than ALC0315-RNA particles following transfection into mouse primary bone marrow dendritic cells [42].

Next, we assessed the potency of each LNP formulation in vivo following injection in BALB/c mice. SM102-RNA and ALC0315-RNA resulted in higher total luciferase signal compared to KC2-RNA. This result is most likely due to the branched ester chains of the SM-102 and ALC-0315 lipid tails, which have been shown to increase functional delivery of mRNA, and not the apparent pKa of the ionizable lipid [43,44]. These results are in accordance with a previous study comparing ALC0315-RNA, SM102-RNA, and MC3-RNA particles administered intravenously into zebrafish embryos, in which ALC0315-RNA and SM102-RNA demonstrated elevated EGFP protein expression compared to MC3-RNA (pKa = 6.4) [42]. In addition, we observed some similarities between the in vitro and in vivo potency, with SM102-RNA and ALC0315-RNA resulting in higher luciferase expression than all other groups in both cases. In comparison, both DNA and RNA formulations of SM-102 and ALC-0315 LNPs demonstrated similar levels of potency and were significantly greater than KC2 LNPs when evaluated in vivo, but not in vitro. Ultimately, the predictive capacity of in vitro screening may be greatly influenced by the cell line and the use of primary cells should be considered to improve in vitro–in vivo translation [27,45,46]. Moreover, this is also the first demonstration that LNPs formulated with SM-102 or ALC-0315 are effective for the delivery of pDNA in vivo.

We then investigated the biodistribution of the luciferase signal following intramuscular injection. In addition to the expected signal surrounding the tibialis anterior, we observed signals in the livers of all mice, except those injected with KC2-DNA, which peaked at six hours. These findings are in line with previous reports that observed localization of LNPs to the liver following intramuscular injection of LNPs formulated with ALC-0315 or SM-102 [30,33]. The uptake of ionizable LNPs by hepatocytes is thought to occur through low-density lipoprotein LDL receptor-mediated endocytosis facilitated by binding to apolipoprotein E [47,48,49]. In the present study, we also found that the injection site signal was strongly correlated with the liver signal for all three LNP formulations. This result indicates that the liver signal is likely proportional to the magnitude of the injection site signal. This is further supported by the fact that KC2-DNA, which resulted in the lowest total luciferase expression of all groups, had no detectable liver signal. To further characterize the biodistribution, we compared the liver-to-injection site ratio among the three formulations, which revealed a significantly higher ratio for the ALC0315-LNPs. This result suggests that the ALC-0315-based particle may have increased hepatic tropism compared to the SM-102-based formulation. This is the first head-to-head comparison of the biodistribution of ALC-0315- and SM-102-based LNPs, and future studies should continue to explore how differences in biodistribution may be beneficial for the treatment or prevention of different diseases.

Finally, we analyzed the immunogenicity of each nanoparticle formulation by quantifying the antibody response to luciferase 28 days after injection. While it is valuable to understand the expression kinetics of these delivery systems, the usefulness of these LNP formulations for mRNA and DNA vaccine delivery is largely determined by their ability to induce a robust antibody response against the encoded antigen [50,51]. Only the SM102-RNA and ALC0315-RNA groups, which had the highest total luciferase expression, induced significant anti-luciferase serum antibody titres when compared to the empty particle control. Antibody responses to both particles were predominated by IgG2a, suggesting a proinflammatory Th1 bias that may be preferential for vaccination against some pathogens [52,53,54]. We found that the antibody titre moderately correlated with both the peak and total luciferase expression suggesting that in vivo bioluminescence imaging may be a viable method for screening novel LNP formulations. In addition to humoral immunity, effective vaccines may also induce a cellular immune response against the desired pathogen. Although T cell responses were not investigated in the present study, a previous comparison of SM102-RNA, AL0315-RNA, and MC3-RNA found that subcutaneous injection of all three particles induced similar levels of intracellular cytokine production by antigen-specific T cells [42]. We suggest that further analyses of the immune mechanisms stimulated by leading LNP formulations are required to better inform the development of effective DNA vaccines.

To our knowledge, our study is the first to demonstrate that LNPs formulated with SM-102 and ALC-0315 are effective for in vivo delivery of pDNA and result in significant and prolonged protein expression. Since both the SM-102 and ALC-0315 LNP formulations have been approved for clinical use, this finding could accelerate the development and approval of DNA vaccines and therapeutics for a variety of diseases. Despite the many advantages of the DNA vaccines, there are inherent obstacles associated with this platform. First, DNA vaccines are typically considered weakly immunogenic and are thought to require an adjuvant for sufficient immune activation [55,56]. However, the use of LNPs for DNA delivery has the potential to overcome this barrier due to their immunostimulatory properties [10,57]. Second, DNA vaccines have been historically associated with possible safety concerns, including integration into the host genome and the development of anti-DNA antibodies [58,59]. More recent investigations into these claims have largely negated these concerns; however, future studies should continue to evaluate the safety of the DNA vaccine platform [22,60,61].

One limitation of our study is the use of the luciferase protein as a surrogate antigen. Luciferase is a relatively weak immunogen, and its use may not accurately reflect the efficacy of other potential vaccine antigens [62]. Therefore, LNP formulations in this study that did not induce a significant antibody response, including all DNA groups, may be immunogenic when encoding a more immunogenic antigen. Furthermore, in this study, only a single dose of each vaccine was administered. Considering that most vaccines require a multi-dose regimen for optimal efficacy, subsequent injections of the surrogate vaccine used in this study, or the inclusion of an adjuvant, may induce a significant antibody response. In fact, a recent study by our group demonstrated that a DNA-KC2 LNP, the poorest performer in our study in terms of in vivo potency and immunogenicity, encoding the outer surface protein C antigen of *B. burgdorferi*, elicited a significant immune response after two 12 ug doses of pDNA, a lower dose than what was administered here [32]. Based on the results of the present study, future studies should explore the immunogenicity and protection conferred by DNA-LNP vaccines formulated with SM-102 or ALC-0315 delivering varying doses of pDNA encoding a relevant antigen.

In conclusion, this study provides important insights into the comparison of three different LNP formulations for RNA and DNA vaccine delivery. We elucidated the potency, expression kinetics, biodistribution, and immunogenicity of some of the most widely used LNP formulations. Specifically, we found that formulations containing the ionizable lipids SM-102 and ALC-0315 delivering an RNA payload were the most potent and immunogenic, while the same lipid formulations delivering DNA resulted in the longest duration of luciferase signal. In addition, the ALC0315-RNA group was also found to have increased hepatic tropism compared to other LNP groups. Our study is the first to demonstrate the utility of LNP formulations identical to those approved for COVID-19 mRNA vaccines for DNA vaccine delivery. Overall, this research could inform the development and optimization of nucleic acid vaccines used for the prevention of a variety of infectious diseases.

## Figures and Tables

**Figure 1 vaccines-11-01580-f001:**
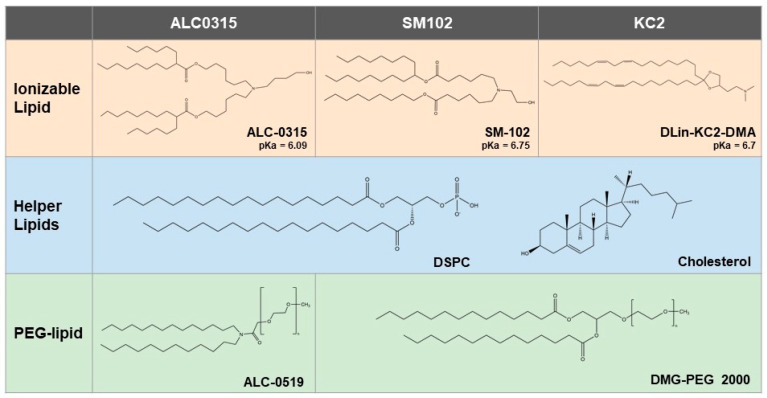
Chemical structures of components in the tested lipid nanoparticle formulations. The header row indicates the particle names referred to in this study.

**Figure 2 vaccines-11-01580-f002:**
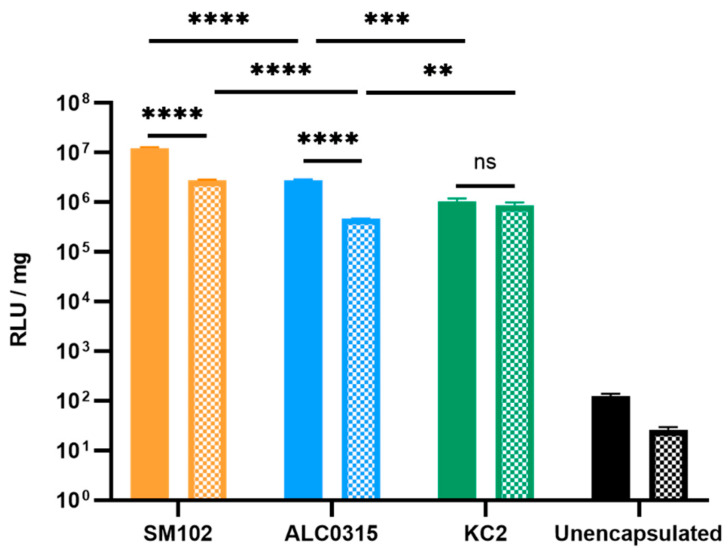
SM102-RNA results in the highest luciferase expression in vitro. HEK293T cells were transfected with 500 ng of mRNA-Luc or DNA-Luc unencapsulated or encapsulated in one of three LNP formulations. Relative luminescence units (RLU) were measured at 16 h for RNA (solid) and 48 h for DNA (stippled) and were standardized per mg of protein quantified by Bicinchoninic Acid (BCA) assay. Error bars represent standard deviation (SD). ns: not significant, ** *p*-value < 0.01, *** *p*-value < 0.001, **** < 0.0001.

**Figure 3 vaccines-11-01580-f003:**
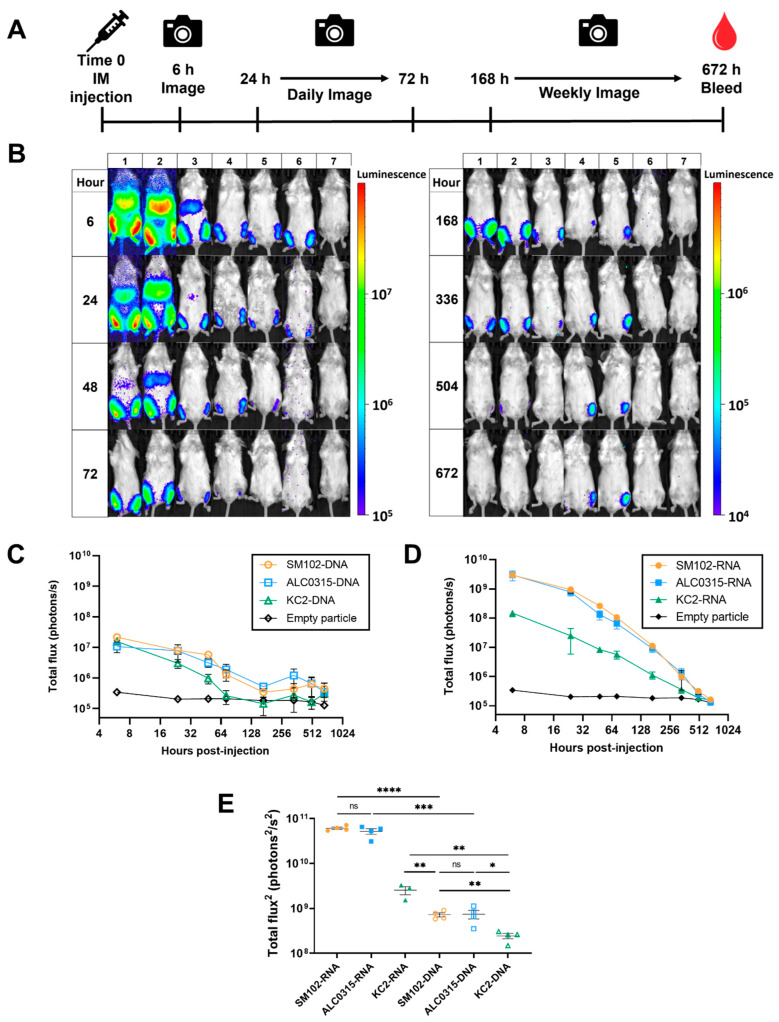
SM102-RNA and ALC0315-RNA result in the highest luciferase expression in vivo. (**A**) Schematic depicting the injection and imaging schedule in BALB/c mice. Mice were injected intramuscularly (IM) with either 1µg of RNA or 25 µg of pDNA encapsulated in one of the LNP formulations. Mice were imaged at 6 h, 24 h, 48 h, 72 h, 168 h, 336 h, 504 h, and 672 h using an IVIS Lumina XRMS imaging system. After the final imaging time point, mice were sacrificed and bled. (**B**) Representative whole-body bioluminescence imaging at different time points after IM administration. (1) SM102-RNA; (2) ALC0315-RNA; (3) KC2-RNA; (4) SM102-DNA; (5) ALC0315-DNA; (6) KC2-DNA; (7) empty particle control. Luciferase signal expression kinetics following intramuscular particle injection. Quantification of whole-body bioluminescent signal over time in BALB/c mice injected with luciferase encoded by (**C**) DNA or (**D**) RNA encapsulated in various particle formulations. (**E**) Total in vivo signal (total flux^2^) produced for each formulation over the complete time-course. Total in vivo signal was calculated as area under the curve (AUC) of the fitted signal curves minus the average AUC of the empty particle control. Error bars represent standard deviation (SD). ns: not significant, * *p*-value < 0.05, ** *p*-value < 0.01, *** *p*-value < 0.001, **** < 0.0001.

**Figure 4 vaccines-11-01580-f004:**
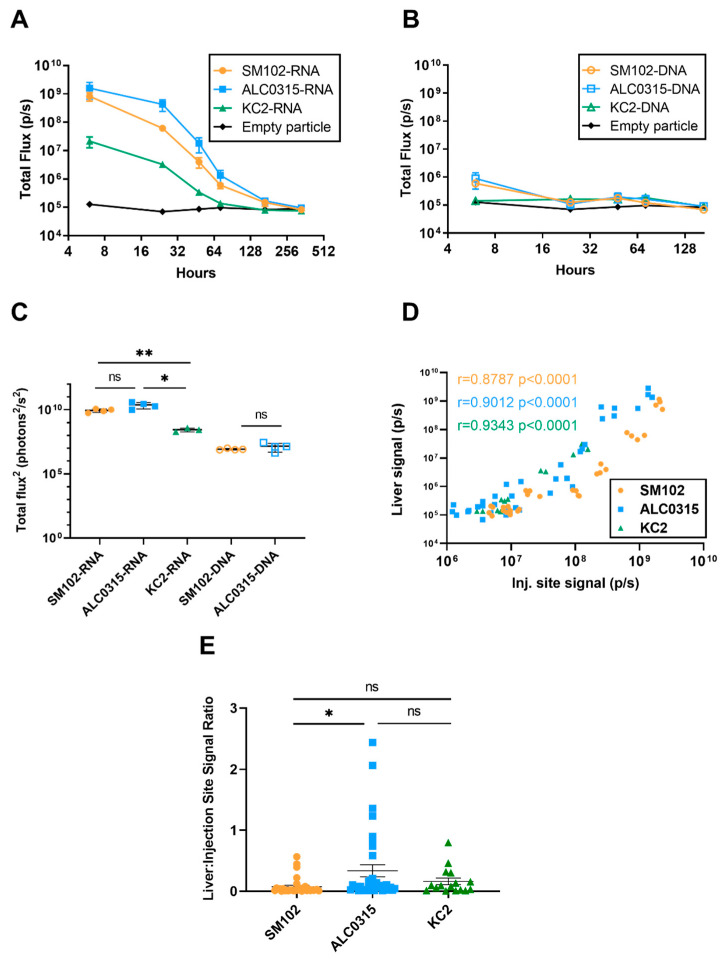
LNPs result in protein expression in the liver proportional to the expression at the injection site. Quantification of liver bioluminescent signal in BALB/c mice injected with (**A**) RNA- or (**B**) DNA-encoding luciferase encapsulated in various particle formulations. (**C**) Total liver signal (total flux^2^) produced for each formulation over the complete time-course. Total liver signal as calculated as area under the curve (AUC) of the fitted liver signal curves minus the average AUC of the empty control group. (**D**) The correlation between liver and injection site signal and (**E**) ratio of liver and injection site signal at time points with detectable liver signal. Error bars represent standard deviation (SD). ns: not significant, * *p*-value < 0.05, ** *p*-value < 0.01.

**Figure 5 vaccines-11-01580-f005:**
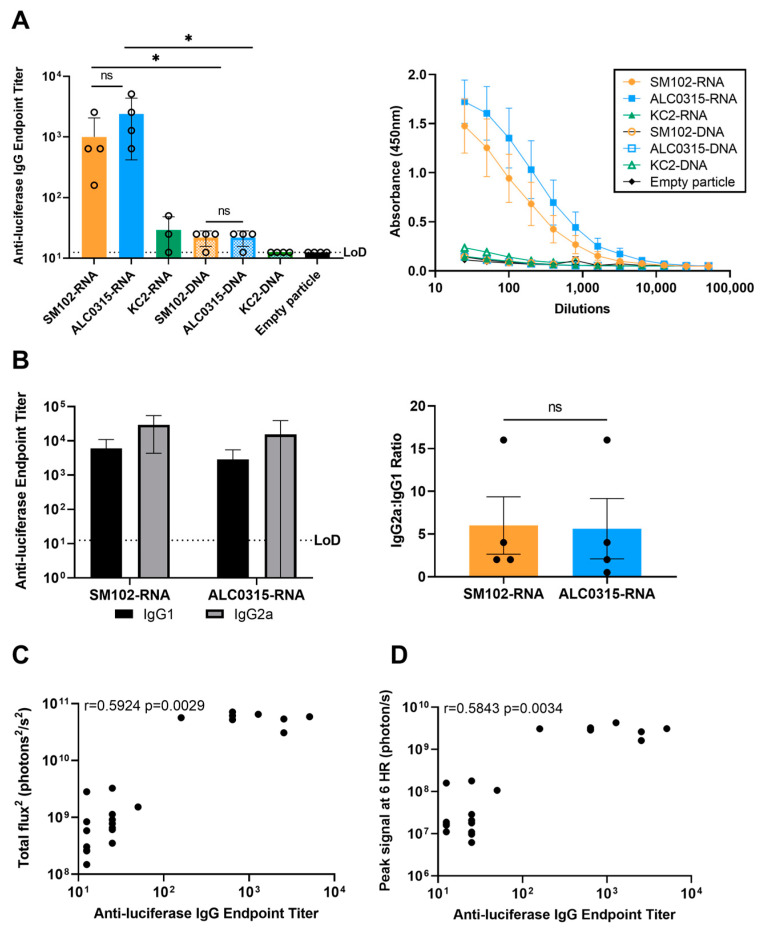
SM102-RNA and ALC0315-RNA particles result in the highest antibody response with Th1 bias. (**A**) ELISA determination of luciferase-specific IgG antibody titres in serum of mice 28 days post-injection. (**B**) IgG2a and IgG1 luciferase-specific serum antibody titres and ratios of IgG2a and IgG1 endpoint titre. Dashed line represents limit of detection (LoD). Error bars represent standard deviation (SD). * *p*-value < 0.05. Correlation between luciferase-specific IgG antibody titres and (**C**) total in vivo signal or (**D**) peak signal at 6 h post-injection.

**Table 1 vaccines-11-01580-t001:** Lipid nanoparticle formulas and characterization. LNP size was determined by nanoparticle tracking analysis and encapsulation efficiency was determined by SYBR™ Gold assay. All particles were made with a polymer amine (N = nitrogen) group to nucleic acid phosphate (P) group (N/P) ratio of 6:1. * Representative of multiple LNP fabrications.

Particle Name	Encapsulated Nucleic Acid	Components	Molar Ratio	LNP Diameter * (Mean ± SD, nm)	Encapsulation Efficiency *
SM102-DNA	DNA	SM-102:DSPC:Chol:DMG-PEG 2000	50:10:38.5:1.5 [30,31]	78 ± 24	97%
ALC0315-DNA	DNA	ALC-0315:DSPC:Chol:ALC-0159	46.3:9.4:42.7:1.6 [33]	67 ± 19	97%
KC2-DNA	DNA	KC2:DSPC:Chol:DMG-PEG 2000	50:10:38.5:1.5 [30,31]	82 ± 36	99%
SM102-RNA	RNA	SM-102:DSPC:Chol:DMG-PEG 2000	50:10:38.5:1.5 [30,31]	74 ± 20	88%
ALC0315-RNA	RNA	ALC-0315:DSPC:Chol:ALC-0159	46.3:9.4:42.7:1.6 [33]	71 ± 21	92%
KC2-RNA	RNA	KC2:DSPC:Chol:DMG-PEG 2000	50:10:38.5:1.5 [30,31]	83 ± 31	97%

**Table 2 vaccines-11-01580-t002:** Luciferase signal duration at the injection site and liver. Signal duration was determined as the last time point where the luminescence was higher than the cut-off value. The cut-off value was defined as the mean of the signal from the empty particle control group plus three times the standard deviation. Duration listed as not applicable (N/A) if all time points had signal below the cut-off value.

Particle Name	Signal Duration (Inj. Site)	Signal Duration (Liver)
SM102-DNA	>28 days	48 h
ALC0315-DNA	>28 days	72 h
KC2-DNA	48 h	N/A
SM102-RNA	21 days	7 days
ALC0315-RNA	21 days	7 days
KC2-RNA	14 days	72 h

## Data Availability

Data are available upon reasonable request to the corresponding author.

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
