# Peer review of "The Expression Kinetics and Immunogenicity of Lipid Nanoparticles Delivering Plasmid DNA and mRNA in Mice"

_vaccines, 2023, doi:10.3390/vaccines11101580_

Round 1

Reviewer 1 Report

The authors compared the potency, expression kinetics, biodistribution, and immunogenicity of DNA-LNP with mRNA-LNP. However, there are some comments should be addressed. Below are specific comments.

1. “delivering DNA and RNA vaccines” in the title is not suitable as both LNP-formulated mRNA and pDNA encode firefly luciferases but not vaccine-relevant antigens.

2. The authors should demonstrate the advantages of DNA-LNP over mRNA-LNP as they claim that the use of DNA-LNP may alleviate poor stability, cold storage requirements, and high production costs.

3. The disadvantages of DNA-LNP vaccines should be discussed since mRNA-LNP vaccines are still in a dominant position.

4. The sequences of pcDNA3-Luc plasmid encoding Luc from Addgene are different from those of mRNA-Luc purchased from TriLink Biotechnologies. Thus, it is hard to compare mRNA and DNA formulated with the same particles.

5. The N/P or w/w ratio of ionizable lipid/mRNA or lipid/DNA should be listed in Table 1. In addition, the error values or PDI of the LNP diameter should be provided.

6. References should be cited to determine the molar ratio in Table 1.

7. Line 255, What is the meaning of third-generation particles?

8. The figure legends of Figure 4 and Figure 5 are the same.

9. It is difficult to conclude that significant luciferase expression was observed in the livers of all mice injected with 290 each of the LNP formulations, with the exception of the KC2-DNA group (line 290) from Figure 3.

10 . The figure legends in Figure 5D, 5E, 6E, and 6F are missing. Besides, these figures are not mentioned in the main text.

NA

Author Response

  1. “delivering DNA and RNA vaccines” in the title is not suitable as both LNP-formulated mRNA and pDNA encode firefly luciferases but not vaccine-relevant antigens.

Thank you for your suggestion. We have modified the title to read:

“Expression kinetics and immunogenicity of lipid nanoparticles delivering plasmid DNA or mRNA in mice”

  1. The authors should demonstrate the advantages of DNA-LNP over mRNA-LNP as they claim that the use of DNA-LNP may alleviate poor stability, cold storage requirements, and high production costs.

Thank you for your comments. We agree that the advantages of DNA-LNP vaccines should be highlighted. Please see the additional text on lines 62-66:

“In comparison, plasmid DNA (pDNA) vaccines are more thermostable and less susceptible to degradation [17,18]. Moreover, DNA vaccines are less expensive to produce, store, and transport than mRNA vaccines [19,20]. LNP-based DNA vaccines therefore have the potential to alleviate a number of issues inherent to mRNA vaccine technology that could improve suitability for more wide-spread use.”

  1. The disadvantages of DNA-LNP vaccines should be discussed since mRNA-LNP vaccines are still in a dominant position.

Thank you for your comments. We agree that the disadvantages of DNA-LNP vaccines should be discussed. Please see the additional text on lines 474-483:

“Despite the many advantages of the DNA vaccines, there are inherent obstacles associated with this platform. First, DNA vaccines are typically considered weakly immunogenic and are thought to require an adjuvant for sufficient immune activation [52,53]. However, the use of LNPs for DNA delivery has the potential to overcome this barrier due to their immunostimulatory properties [10,54]. Second, DNA vaccines have been historically associated with possible safety concerns, including integration into the host genome and the development of anti-DNA antibodies [55,56]. More recent investigations into these claims have largely negated these concerns, however, future studies should continue to evaluate the safety of the DNA vaccine platform [19,57,58].”

  1. The sequences of pcDNA3-Luc plasmid encoding Luc from Addgene are different from those of mRNA-Luc purchased from TriLink Biotechnologies. Thus, it is hard to compare mRNA and DNA formulated with the same particles.

Thank you for your comment. Off-the-shelf products were purchased for this study to reduce the overall cost. Although the sequences of the pcDNA3-Luc purchase from Addgene are different from those of mRNA-Luc purchased from TriLink,  both sequences were optimized for mammalian expression. In addition, in a 2023 study by Takanashi et al, the authors compared protein expressions after LNP-mRNA or LNP-DNA injection, encoding the same sequence. Nevertheless, their findings echo our results, observing a higher expression from the LNP-mRNA group. Therefore, we think that the differences in the sequences are not detrimental to the comparison.

  1. The N/P or w/w ratio of ionizable lipid/mRNA or lipid/DNA should be listed in Table 1. In addition, the error values or PDI of the LNP diameter should be provided.

Thank you for your comment. The N/P ratio is now added to the Table 1 caption and method section 2.2. Lipid nanoparticle (LNP) generation. All particles were made with an N/P ratio of 6:1.

  1. References should be cited to determine the molar ratio in Table 1.

Thank you for your comment. References are now added to the molar ratios in Table 1.

  1. Line 255, What is the meaning of third-generation particles?

We apologize for the confusion. SM102 and ALC0315 are sometimes referred to as “third-generation particles”. In general, lipid nanoparticles are classified into three generations based on their properties. The first generation particles have moderate transfection efficiency but with notable toxicity due to the non-biodegradable nature of it. The second generation particles, such as the KC2 that was tested in this study, are biodegradable with the addition of ester linkers. With the improved delivery efficiency, this led to the approval of Onpattro, which had MC3 as their main component. The third generation particles, such as SM102 and ALC-0315, are further improved based on the second generation, which enables them to deliver longer mRNAs and be widely used for the COVID vaccines.

To avoid confusion, we have modified the result section to remove the use of “third-generation”:

Line 271-272: “The peak intensity of KC2-RNA particles was approximately 20-fold lower than ALC0315-RNA (p=0.002) and SM102-RNA (p=0.0021).”

  1. The figure legends of Figure 4 and Figure 5 are the same.

Thank you for bringing this error to our attention. The caption for the original figure 5, now numbered figure 4, now reads:

Figure 4. LNPs result in protein expression in the liver proportional to the expression at the injection site. Quantification of liver bioluminescent signal in BALB/c mice injected with A) RNA or B) DNA encoding luciferase encapsulated in various particle formulations. C) Total liver signal (total flux2) produced for each formulation over the complete time-course. Total liver signal as calculated as area under the curve (AUC) of the fitted liver signal curves minus the average AUC of the empty control group. D) The correlation between liver and injection site signal and E) ratio of liver and injection site signal at time points with detectable liver signal. Error bars represent standard deviation (SD). ns: not significant, * p-value < 0.05, ** p-value < 0.01.”

  1. It is difficult to conclude that significant luciferase expression was observed in the livers of all mice injected with each of the LNP formulations, with the exception of the KC2-DNA group (line 290) from Figure 3.

Thank you for your comment. The scale bars in figure 3 were originally adjusted to provide the clearest representation of the maximum signal intensity without overexposure. As a result, low-intensity signals, such as signals from the liver, are not visible in the representative figure. However, such signals were still quantified and have been depicted in figures 3C-E. Thus, quantitative data was used to justify our statements rather than the representative images.

To determine whether the luciferase expression was significant at a timepoint, the cut-off value was defined as the mean of the signal from the empty particle group plus three times the standard deviation. From this calculation, it was found that only the KC2-DNA group did not have significant luciferase expression at any of the timepoints.

For clarification, a line was added to the results:

Line 324-326: “The cut-off value for significant luciferase expression was defined as the mean of the signal from the empty particle group plus three times the standard deviation.”

This calculation is also described in the method section 2.7 Mathematical and statistical analyses:

Line 204-205: “The cut-off value for in vivo signal duration was defined as the mean of the signal from the empty particle group plus three times the standard deviation.”

10 . The figure legends in Figure 5D, 5E, 6E, and 6F are missing. Besides, these figures are not mentioned in the main text.

Thank you for bring this copy-paste and format error to our attention. Figures 5 and 6, now numbered Figures 4 and 5 respectively, have been edited for clarity and their captions were corrected.

Reviewer 2 Report

Comments to the Authors:

This manuscript described three different lipid nanoparticle (LNP) formulations for RNA and DNA vaccine delivery, and compared three leading LNP formulations based on potency, expression kinetics, biodistribution, and immunogenicity. This study demonstrated that the LNP formulation was identical to the COVID-19 mRNA vaccine that have been approved for  DNA vaccine delivery. This is a very important and meaningful work, which informs the development and optimization of nucleic acid vaccines for the prevention of a variety of infectious diseases. This manuscript can be published in Vaccines, however, some revisions are needed before this article can be published.

1.       Carefully check the language in the paper for standardization, including the use of abbreviations, and grammatical issues. The formatting in the text is not uniform, and there are also many formatting errors. For example, page 3, line 125, “CO2” should be “CO2”; page 3, line 125, “37°C” should be “37 °C”; page 3, line 126 and 129, “µL” should be “µL”.

2.       The text size in Figures 1 and 3 is too small. Please improve the clarity of the picture and modify the text in the picture to the appropriate size.

3.       The format of references is not uniform. For example, Ref. 8, change “2877-2892” to “2877–2892”; Ref. 10, change “4414–8” to “4414–4418”; Ref. 31, change “1055–65” to “1055–1065”. In addition, more related papers and the latest research are needed to be cited.

4.       The author stated that “SM102-RNA particles result in the highest luciferase expression in vitro”. However, the authors did not give a detailed analysis of this phenomenon. Please analyze in detail why SM102-RNA particles result in the highest luciferase expression in vitro.

5.       Introduction, the author mentions “Second, LNPs improve cellular uptake leading to increased expression of the target antigen, which may contribute to increased immunogenicity”. In order to support this statement, the following recently published important related papers should be cited: Exploration 2021, 1, 21; Sci. China: Chem. 2023, 66, 613; VIEW. 2023;20220064; Adv. Mater. 2023, DOI: 10.1002/adma.202304249.

Moderate editing of English language required

Author Response

1.       Carefully check the language in the paper for standardization, including the use of abbreviations, and grammatical issues. The formatting in the text is not uniform, and there are also many formatting errors. For example, page 3, line 125, “CO2” should be “CO2”; page 3, line 125, “37°C” should be “37 °C”; page 3, line 126 and 129, “µL” should be “µL”.

Thank you for your suggestion. We have made the requested changes for “CO2”, “37 °C”, and  “µL” at all places in the manuscript. We have also standardized the formatting and revised the manuscript for any grammatical issues.

  1. The text size in Figures 1 and 3 is too small. Please improve the clarity of the picture and modify the text in the picture to the appropriate size.

Thank you for your comments. The text size has been adjusted in Figures 1 and 3, and modifications were made to improve the clarity of the figures.

  1. The format of references is not uniform. For example, Ref. 8, change “2877-2892” to “2877–2892”; Ref. 10, change “4414–8” to “4414–4418”; Ref. 31, change “1055–65” to “1055–1065”. In addition, more related papers and the latest research are needed to be cited.

Thank you for your suggestion. We have modified all format of the references to be uniform. In addition, we added several more references throughout the text to reflect the latest publications in the field.

  1. The author stated that “SM102-RNA particles result in the highest luciferase expression in vitro”. However, the authors did not give a detailed analysis of this phenomenon. Please analyze in detail why SM102-RNA particles result in the highest luciferase expression in vitro.

Thank you for this comment. We have added the following in the result and discussion sections:

Lines 240-241: “For both RNA- and DNA-LNPs, the SM-102 formulation resulted in the highest luciferase expression… likely due to the unique structural characteristics of SM-102.”

This observation has been discussed further in the discussion:

Lines 401-411: “...The difference in reporter gene expression between SM102-DNA and KC2-DNA LNPs is likely attributed to SM-102 disrupting the endosomal membrane to a greater extent due to its more cone-like shape [26]. Compared to ALC-0315 and KC2, SM-102 has a more asymmetric tail, which has been shown to have higher transfection efficiency [36]. Interestingly, for RNA-LNPs a different pattern was observed with SM102-RNA > ALC0315-RNA > KC2-RNA suggesting that factors other than pKa, such as LNP morphology, inner structure, and RNA microenvironment, may account for the observed differences. In a recent publication, the authors observed several lipid structure activity relationships that correlated with improved protein expression, including number of carbons on the lipid tails on the ester side and the effect of carbon spacing on the disulfide arm of the lipids [38].

  1. Introduction, the author mentions “Second, LNPs improve cellular uptake leading to increased expression of the target antigen, which may contribute to increased immunogenicity”. In order to support this statement, the following recently published important related papers should be cited: Exploration 2021, 1, 21; Sci. China: Chem. 2023, 66, 613; VIEW. 2023;20220064; Adv. Mater. 2023, DOI: 10.1002/adma.202304249.

Thank you for providing us with related journals. We have added VIEW. 2023;20220064 and Adv. Mater. 2023 to our references. Both reviews highlighted the central role nanoparticles play  in improving cellular uptake.

Reviewer 3 Report

  1. Abstract:
  • The abstract lacks a clear statement of the research objective, and it should provide a concise overview of the study's main findings.
  • It does not include any statistical information (e.g., p-values) to support the claims made about the potency, expression kinetics, biodistribution, and immunogenicity of the tested LNP formulations.
  1. Introduction:
  • The introduction should provide a more comprehensive background on the topic, particularly regarding DNA vaccine delivery using LNPs. The current introduction is too focused on mRNA vaccines and their success during the COVID-19 pandemic.
  • It lacks citations for specific studies when discussing the limitations of mRNA vaccines (e.g., poor stability, cold storage requirements, and high production costs).
  • There is no clear statement of the research gap or objective of the study, which should be presented at the end of the introduction.
  1. Materials and Methods:
  • The section is too brief and lacks sufficient details about the experimental procedures and methodologies used. Specific information about the dosages, administration protocols, and control groups should be included.
  • The statistical analysis methods need to be described more thoroughly, including how the data were analyzed and the significance thresholds used for comparisons.
  1. Results:
  • The results section is poorly organized. It should be divided into subsections for each research objective, such as potency, expression kinetics, biodistribution, and immunogenicity, to facilitate better understanding and interpretation of the data.
  • Figures and tables should be provided to visually represent the findings, making it easier for readers to grasp the results.
  1. Discussion:
  • The discussion section should provide a comprehensive interpretation of the results in light of the existing literature. It should compare the findings with previous studies on LNP-based DNA and mRNA vaccines, particularly addressing the unique aspects of the current study.
  • The manuscript should explore the implications of the findings, potential applications of the tested LNP formulations, and future research directions.
  • The limitations of the study and potential sources of bias or error should be acknowledged and discussed.
  1. Conclusion:
  • The conclusion should be a concise summary of the key findings and their significance.
  • It should also reiterate the research gap and highlight the novel contributions of the study.
  • The manuscript needs to be reorganized with clear section headings to improve readability.
  • The text should be thoroughly proofread to correct grammatical errors and improve the overall writing style.

    Ethical Considerations:
  • The manuscript lacks a statement about ethical approval for animal experiments and compliance with relevant guidelines. It is essential to provide ethical considerations and confirm adherence to the principles of animal research.

To enhance the overall quality of the manuscript, it would be beneficial to have a proofreader or language editor review the text for grammatical and linguistic improvements. This would help ensure that the language is clear, precise, and communicates the findings effectively to the target audience.

Author Response

1. Abstract:

  • The abstract lacks a clear statement of the research objective, and it should provide a concise overview of the study's main findings.
  • It does not include any statistical information (e.g., p-values) to support the claims made about the potency, expression kinetics, biodistribution, and immunogenicity of the tested LNP formulations.

Thank you for your suggestions. We agree that the abstract would benefit from these additions and we have also included the relevant statistical information. A clear statement of the research objective can be found on lines 30-32. The concise overview of the study’s main findings can be found on lines 35-40 of the revised manuscript. The revised abstract now reads:

Abstract: In recent years, lipid nanoparticles (LNPs) have emerged as a revolutionary technology for vaccine delivery. LNPs serve as an integral component of mRNA vaccines by protecting and transporting the mRNA payload into host cells. Despite their prominence in mRNA vaccines, there remains a notable gap in our understanding of the potential application of LNPs for the delivery of DNA vaccines. In this study, we sought to investigate the suitability of leading LNP formulations for the delivery of plasmid DNA. In addition, we aimed to explore key differences in the properties of leading LNP formulations when delivering either mRNA or DNA. To address these questions, we compared three leading LNP formulations encapsulating mRNA or pDNA encoding firefly luciferase based on potency, expression kinetics, biodistribution, and immunogenicity. Following intramuscular injection in mice, we determined that RNA-LNPs formulated with either SM-102 or ALC-0315 lipids were the most potent (all p-values < 0.01) and immunogenic (all p-values < 0.05), while DNA-LNPs formulated with SM-102 or ALC-0315 demonstrated the longest duration of signal. Additionally, all LNP formulations were found to induce expression in the liver that was proportional to the signal at the injection site (SM102: r=0.8787, p<0.0001; ALC0315: r=0.9012, p<0.0001; KC2: r=0.9343, p<0.0001). Overall, this study provides important insights into the differences between leading LNP formulations and their applicability to DNA- and RNA-based vaccinations.”

2. Introduction: The introduction should provide a more comprehensive background on the topic, particularly regarding DNA vaccine delivery using LNPs. The current introduction is too focused on mRNA vaccines and their success during the COVID-19 pandemic.  It lacks citations for specific studies when discussing the limitations of mRNA vaccines (e.g., poor stability, cold storage requirements, and high production costs). There is no clear statement of the research gap or objective of the study, which should be presented at the end of the introduction.

Thank you for your suggestions. We have modified the introduction to include more information on LNP delivery of DNA vaccines. We have also included references to support the claims regarding the limitations of mRNA vaccines. Please see the revised section of the introduction below:

Lines 59-80: “To date, all approved LNP-based vaccines have been designed to encapsulate modified mRNA. Despite the success of this technology, there are several limitations associated with mRNA vaccines, including poor stability, cold storage requirements, and high production costs [13-16]. In comparison, plasmid DNA (pDNA) vaccines are more thermostable and less susceptible to degradation [17,18]. Moreover, DNA vaccines are less expensive to produce, store, and transport than mRNA vaccines [19,20]. LNP-based DNA vaccines therefore have the potential to alleviate a number of issues inherent to mRNA vaccine technology that could improve suitability for more wide-spread use.

Recently, the first DNA vaccine was approved in India for the prevention of COVID-19 [21]. Many other DNA vaccines in clinical and preclinical development are administered using intradermal injection or specialized instruments such as gene guns or needle-free injectors [22,23]. Considering that these techniques may not be easily translated into human vaccination programs, LNPs offer a safe and reliable alternative for delivering DNA vaccines by intramuscular injection [6,24,25]. Despite these advantages, there are currently no DNA vaccines approved or undergoing clinical trials that are delivered by LNPs. One study by Mucker and colleagues demonstrated that LNP encapsulation increased the neutralizing anti-body titres induced by DNA vaccines for Andes virus and Zika virus [6]. In addition, a study by Algarni et al. demonstrated that a DNA-LNP vaccine formulated with the ionizable lipid DLin-KC2-DMA (KC2), resulted in greater antigen expression than the leading DLin-MC3-DMA (MC3)-formulated particle when administered intramuscularly [26]. These studies provide promising insights into the potential of DNA-LNP vaccines as an alternative platform for vaccine development.”

In addition, we have highlighted the research gap and objective of the study as follows:

Lines 81-86: “To our knowledge the potency, expression kinetics, biodistribution, and immunogenicity conferred by DNA-LNP vaccines formulated with the ionizable lipid KC2 or the lipid formulations utilized in clinical COVID-19 vaccines (SM-102 and ALC-0315) have not yet been investigated [27,28]. This study aims to bridge a significant knowledge gap regarding the efficacy and characteristics of DNA-LNP vaccines formulated with different lipid components.”

3. Materials and Methods: The section is too brief and lacks sufficient details about the experimental procedures and methodologies used. Specific information about the dosages, administration protocols, and control groups should be included. The statistical analysis methods need to be described more thoroughly, including how the data were analyzed and the significance thresholds used for comparisons

Thank you for your suggestions and we apologize for any confusion regarding the details of the experiments. We have combined section 2.1 and 2.6 to improve the clarity. The information about the dosages, administration protocols, and control groups can be found on lines 163-170:

“Mice were randomly divided into eight groups, with three to four in each group. At time zero, mice received intramuscular injection into the left and right tibialis anterior muscle, with a total injection volume of 50 µL. The injection contained either 1 µg of mRNA or 25 µg of pDNA encapsulated in LNPs diluted in PBS. An LNP containing no nucleic acid (empty particle) control group was also included in the study and dosed with the equivalent amount of lipid as the DNA-LNP groups. Depilatory cream was applied to the mouse legs under anesthesia to improve visualization prior to initial injection and again 17 days after injection.”

The significance thresholds and additional information have also been included in the statistical analysis section on lines 203, 208, 210, 214.

4. Results: The results section is poorly organized. It should be divided into subsections for each research objective, such as potency, expression kinetics, biodistribution, and immunogenicity, to facilitate better understanding and interpretation of the data.

  • Figures and tables should be provided to visually represent the findings, making it easier for readers to grasp the results

Thank you for your suggestions regarding the presentation of the results. The results have been rearranged to include separate sections for potency, expression kinetics, biodistribution and immunogenicity. Mainly, section 3.2 now focuses solely on the in vitro and in vivo potency while the expression kinetics information has been moved to the new section 3.3 and supplemented with additional information. In conjunction with these changes and to improve interpretation of the results, we have also combined Figure 3 and 4 and we have moved Table 2 to be in line with the expression kinetics results section.

New section 3.3 (Lines 305-315):

“3.3 Expression kinetics: SM102-DNA and ALC0315-DNA particles result in the longest duration of protein expression

Following the peak luciferase expression observed at the six hour time point, protein expression steadily decreased over the remaining time course for all groups. The KC2-DNA formulation resulted in signal significantly above baseline for the shortest duration of all groups, lasting only 48 hours. In comparison, the SM102-DNA and ALC0315-DNA groups were found to have detectable luciferase signal at the final time point of 28 days (Figure 4A, Table 2). SM102-RNA and ALC0315-RNA groups yielded signal for 21 days, while the KC2-RNA formulation lasted a shorter duration of 14 days. Overall, the SM102-DNA and ALC0315-DNA particles resulted in the longest duration of protein expression when compared to all other groups.”

5. Discussion: The discussion section should provide a comprehensive interpretation of the results in light of the existing literature. It should compare the findings with previous studies on LNP-based DNA and mRNA vaccines, particularly addressing the unique aspects of the current study.

  • The manuscript should explore the implications of the findings, potential applications of the tested LNP formulations, and future research directions.
  • The limitations of the study and potential sources of bias or error should be acknowledged and discussed.

Thank you for your recommendations. The discussion has been thoroughly modified to reflect your suggestions.

Additional text discussing existing literature has been added on the following lines:

Line 403: “Compared to ALC-0315 and KC2, SM-102 has a more asymmetric tail, which has been shown to have higher transfection efficiency [36]”

Line 408-414: “In another recent publication, the authors observed several lipid structure activity relationships that correlated with improved protein expression, including number of carbons on the lipid tails on the ester side and the effect of carbon spacing on the disulfide arm of the lipids [38]. The results of our study are consistent with a recent publication by Escalona-Rayo et al. that demonstrated that SM102-RNA particles resulted in greater in vitro protein expression than ALC0315-RNA particles following transfection into mouse primary bone marrow dendritic cells [39].”

Lines 419-423: “These results are in accordance with a previous study comparing ALC0315-RNA, SM102-RNA, and MC3-RNA particles administered intravenously into zebrafish embryos, in which ALC0315-RNA and SM102-RNA demonstrated elevated EGFP protein expression compared to MC3-RNA (pKa = 6.4) [39].”

Lines 462-469: “In addition to humoral immunity, effective vaccines may also induce a cellular immune response against the desired pathogen. Although T cell responses were not investigated in the present study, a previous comparison of SM102-RNA, AL0315-RNA, MC3-RNA, found that subcutaneous injection of all three particles induced similar levels of intra-cellular cytokine production by antigen-specific T cells [39]. We suggest that further analyses of the immune mechanisms stimulated by leading LNP formulations are required to better inform the development of effective DNA vaccines.”

Newly added information about the novelty of the results can be found on…

Lines 394: “To our knowledge, this is the first demonstration that SM-102- and ALC-0315-based LNPs are effective for in vitro transfection of pDNA.”

Lines 430: “Moreover, this is also the first demonstration that LNPs formulated with SM-102 or ALC-0315 are effective for the delivery of pDNA in vivo.”

Lines 447-450 “This is the first head-to-head comparison of the biodistribution of ALC-0315- and SM-102-based LNPs and future studies should continue to explore how differences in biodistribution may be beneficial for the treatment or prevention of different diseases.”

Lines 470-474: “To our knowledge, our study is the first to demonstrate that LNPs formulated with SM-102 and ALC-0315 are effective for in vivo delivery of pDNA and result in significant and prolonged protein expression. Since both the SM-102 and ALC-0315 LNP formulations have been approved for clinical use, this finding could accelerate the development and approval of DNA vaccines and therapeutics for a variety of diseases.”

Implications of the results and future directions can be found on…

Lines 482:“…future studies should continue to evaluate the safety of the DNA vaccine platform [19,57,58].”

Lines 496-498: “Based on the results of the present study, future studies should explore the immunogenicity and protection conferred by DNA-LNP vaccines formulated with SM-102 or ALC-0315 delivering varying doses of pDNA encoding a relevant antigen.”

Limitations of the study are discussed in the second-to-last paragraph of the manuscript and further information on limitations of the study have been added on…

Lines 474-483: “Despite the many advantages of the DNA vaccines, there are inherent obstacles associated with this platform. First, DNA vaccines are typically considered weakly immunogenic and are thought to require an adjuvant for sufficient immune activation [52,53]. However, the use of LNPs for DNA delivery has the potential to overcome this barrier due to their immunostimulatory properties [10,54]. Second, DNA vaccines have been historically associated with possible safety concerns, including integration into the host genome and the development of anti-DNA antibodies [55,56]. More recent investigations into these claims have largely negated these concerns, however, future studies should continue to evaluate the safety of the DNA vaccine platform [19,57,58].”

6. Conclusion: The conclusion should be a concise summary of the key findings and their significance. It should also reiterate the research gap and highlight the novel contributions of the study. The manuscript needs to be reorganized with clear section headings to improve readability. The text should be thoroughly proofread to correct grammatical errors and improve the overall writing style.

Thank you for your suggestions. We have now included a brief summary of the key findings, the novelty of the study, and the significance in the work in the concluding paragraph of the discussion. Please see the revised paragraph below:

“In conclusion, this study provides important insights into the comparison of three different LNP formulations for RNA and DNA vaccine delivery. We elucidated the potency, expression kinetics, biodistribution, and immunogenicity of some of the most widely used LNP formulations. Specifically, we found that formulations containing the ionizable lipids SM-102 and ALC-0315 delivering an RNA payload were the most potent and immunogenic, while the same lipid formulations delivering DNA resulted in the longest duration of luciferase signal. In addition, the ALC0315-RNA group was also found to have increased hepatic tropism compared to other LNP groups. Our study is the first to demonstrate the utility of LNP formulations identical to those approved for COVID-19 mRNA vaccines for DNA vaccine delivery. Overall, this research could inform the development and optimization of nucleic acid vaccines used for the prevention of a variety of infectious diseases.”

Furthermore, we have reorganized the headings as described in comment 4 and revised the grammar and writing style throughout the manuscript.

7. Ethical Considerations: The manuscript lacks a statement about ethical approval for animal experiments and compliance with relevant guidelines. It is essential to provide ethical considerations and confirm adherence to the principles of animal research.

Thank you for your comment. The following sentence is listed in the materials and methods section “2.5 Animal Study”:

Lines 160-162: “All animal procedures were performed in accordance with institutional guidelines and ethical approval was granted by the Animal Care Committee at Health Canada, Ottawa, ON.”

The required institutional review board statement is also included at the end of the manuscript:

Line 522-523: “Institutional Review Board Statement: The animal study protocol was approved by the Institutional Animal Care Committee of Health Canada (protocol #2021-009, approved on October 6, 2021)”

Reviewer 4 Report

I thank the authors for an interesting study. The authors investigated RNA and DNA LNPs formulations containing ionizable lipids SM-102, ALC-0315 or KC2. They elucidated the potency, expression kinetics, biodistribution, and immunogenicity of LNP formulations. The design of the study is not objectionable. Methods are described adequately. The obtained results are clearly described.

This study provides important insights of three different LNP formulations for RNA and DNA vaccine delivery.

There are a number of minorl notes, which are listed below:

1. In Table 1 please provide the standard deviation.

2. Descriptions of DNA and RNA LNPs are provided in the text, but Figure 2 only shows data for RNA LNPs. Data for DNA-LNPs should be shown in the figure.

3. In Fig 3. “A”, “B” are missing. The font for the bioluminescence scale is very small.

4. Please add IgG1 and IgG2a detection to the Methods.

5. Please check the references in Figure 6B-F. Figure 6 caption is not correct

Author Response

  1. In Table 1 please provide the standard deviation.

Thank you for the suggestion. Table 1 is now modified to include the standard deviation for the sizing.

  1. Descriptions of DNA and RNA LNPs are provided in the text, but Figure 2 only shows data for RNA LNPs. Data for DNA-LNPs should be shown in the figure.

Thank you for your suggestion. We apologize for any confusion. As indicated by the figure caption, the data for the DNA-LNPs is shown in Figure 2 by the stippled bars.

  1. In Fig 3. “A”, “B” are missing. The font for the bioluminescence scale is very small.

Thank you for the comment. Letters were added to figure 3 to distinguish each section and the font in the bioluminescence scale was enlarged.

  1. Please add IgG1 and IgG2a detection to the Methods.

We apologize for the confusion. Both IgG1 and IgG2a levels were detected by the ELISA protocol outlined in the Methods section. Please see lines 189-191:

“ ...then HRP-conjugated goat anti-mouse IgG (Cytiva, Marlborough, MA), HRP-conjugated goat anti-mouse IgG1 (Jackson Immuno Research), or HRP-conjugated goat anti-mouse IgG2a (Jackson Immuno Research) were added at a 1:2000 dilution and incubated for one hour at 37 °C.”

  1. Please check the references in Figure 6B-F. Figure 6 caption is not correct

Thank you for your comments. Figure 6 and the corresponding caption were corrected.

Round 2

Reviewer 1 Report

The copy-paste error is a big error. In addition, the authors should demonstrate (a side-by-side comparison) rather than highlight the advantages of DNA-LNP over mRNA-LNP as they claim that (1) DNA vaccines are more thermostable and less susceptible to degradation; (2) the use of DNA-LNP may alleviate poor stability, cold storage requirements, and high production costs.

NA.

Author Response

Our sincerest apologies for the oversight that occurred in the figure caption of the manuscript. We regret any misunderstanding it may have caused. We are implementing more thorough proofreading and review processes for future submissions. Thank you for your understanding.

Thank you for your comment. We agree that comparative studies pertaining to the stability of DNA-LNP and mRNA-LNP would be valuable to the field. We have other on-going projects addressing this particular aspect, including stability under various storage conditions. After careful consideration, we have opted to present the stability study as an independent manuscript due to the large amount of data and the complexity of the experimental design. Therefore, these studies are beyond the scope of the current publication. 

Nevertheless, our statement in the manuscript regarding the stability of the DNA vaccine was made based on the references cited, as well as the current stability profile of approved or clinically tested DNA and mRNA vaccines. For example, ZyCoV-D, a DNA plasmid-based COVID-19 vaccine with no LNP component, is stable at 2–8°C and at room temperature for 3 months. In another open vial study, the vaccine of interest was stable and sterile up to 28 days (https://doi.org/10.1016%2FS0140-6736(22)00151-9). Another COVID-19 DNA vaccine that is currently in phase 3 clinical trials, INO-4800, was stable at both 2–8°C and 25 ± 2°C for more than 6 months, and it was concluded that this vaccine does not require ultra-cold storage (https://doi.org/10.1093/ofid/ofac492.1760) . On the other hand, popular mRNA vaccines encapsulated in LNP have more stringent storage conditions. The Moderna vaccine can be stored in the frozen state at -20 °C for up to 6 months, 30 days at 2–8°C, and only up to 12 h at room temperature. The Pfizer COVID-19 vaccine has a similar stability profile: -80 to -60 °C for up to 6 months, 5 days at 2–8°C, and only up to 2h at room temperature (https://doi.org/10.1016/j.xphs.2020.12.006). 

Finally, in a study by Kafetzis et al., the authors demonstrated that when encapsulated by the same LNP formulation, mRNA are more sensitive to the storage condition. The LNP-mRNA group lost almost all potency when stored at room temperature or 4°C, after just 2 weeks. Whereas the LNP-DNA group was stable for 2 weeks (https://doi.org/10.1002/adhm.202203022). 

We have added these additional references to the manuscript. Thank you.

Reviewer 3 Report

The manuscript has been improved for the publication.

Author Response

Thank you for your review.

Round 3

Reviewer 1 Report

The authors should compare the thermostability of DNA-LNP with that of mRNA-LNP using three types of formulations described in the manuscript.

NA

Author Response

Thank you for your suggestion. As recommended, we have compared the thermostability of the DNA-LNPs and mRNA-LNPs using all three formulations described in the manuscript. In this thermostability test, we compared the in vitro transfection potency of LNPs stored at 37°C for 7 days to the transfection potency after fresh preparation on day 0. We confirmed that across all formulations, the DNA-LNPs maintained potency better than the mRNA-LNPs, which saw significant decreases in luminescence after storage. This data has been included as a supplementary figure (Figure S1) and the following sentence has been included on line 63-65:

“We have observed that DNA-LNPs maintain transfection potency better than mRNA-LNPs after one week at 37°C (Figure S1).”

The figure caption for Figure S1 is as follows:

“Figure S1. DNA-LNPs are more thermostable than RNA-LNPs. LNPs encapsulating mRNA encoding firefly luciferase were stored at 37°C for seven days and then used to transfect HEK293T cells according to the in vitro transfection assay protocol described in the methods section. % change in RLU/mg represents the change in RLU/mg when compared to that of the freshly prepared LNP formulation on day 0. RLU: relative luminescence units. Error bars represent standard deviation (SD). * p-value < 0.05, **** < 0.0001.”